# Nanosurface Texturing for Enhancing the Antibacterial Effect of Biodegradable Metal Zinc: Surface Modifications

**DOI:** 10.3390/nano13132022

**Published:** 2023-07-07

**Authors:** Enmao Xiang, Corey S. Moran, Sašo Ivanovski, Abdalla Abdal-hay

**Affiliations:** 1School of Dentistry, The University of Queensland, Brisbane 4006, Australia; uqexiang@uq.edu.au (E.X.); corey.moran@uq.edu.au (C.S.M.); 2Centre for Orofacial Regeneration, Reconstruction and Rehabilitation (COR3), School of Dentistry, The University of Queensland, Brisbane 4006, Australia; 3Department of Engineering Materials and Mechanical Design, Faculty of Engineering, South Valley University, Qena 85325, Egypt; 4Faculty of Industry and Energy Technology, Mechatronics Technology Program, New Cairo Technological University, Fifth Settlement, Cairo 11835, Egypt

**Keywords:** anti-bacterial, biodegradable Zn, infection, interfacial morphology

## Abstract

Zinc (Zn) as a biodegradable metal has attracted research interest for bone reconstruction, with the aim of eliminating the need for a second removal surgery and minimizing the implant-to-bone transfer of stress-shielding to maintain bone regeneration. In addition, Zn has been shown to have antibacterial properties, particularly against Gram-negative bacteria, and is often used as a surface coating to inhibit bacterial growth and biofilm formation. However, the antibacterial property of Zn is still suboptimal in part due to low Zn ion release during degradation that has to be further improved in order to meet clinical requirements. This work aims to perform an innovative one-step surface modification using a nitric acid treatment to accelerate Zn ion release by increasing surface roughness, thereby endowing an effective antimicrobial property and biofilm formation inhibition. The antibacterial performance against Staphylococci aureus was evaluated by assessing biofilm formation and adhesion using quantitative assays. The surface roughness of acid-treated Zn (Ra ~ 30 nm) was significantly higher than polished Zn (Ra ~ 3 nm) and corresponded with the marked inhibition of bacterial biofilm, and this is likely due to the increased surface contact area and Zn ion accumulation. Overall, surface modification due to nitric acid etching appears to be an effective technique that can produce unique morphological surface structures and enhance the antibacterial properties of biodegradable Zn-based materials, thus increasing the translation potential toward multiple biomedical applications.

## 1. Introduction

Pelvic and femoral bone-related fractures induced by osteoporosis, one of the most common skeletal diseases, is a significant problem affecting the quality of life and mental health in the geriatric population globally [1,2]. Traditionally, medical devices (implants) for the treatment and repair of bone fractures are fabricated from metallic-based non-degradable materials such as titanium, stainless steel, and cobalt–chromium alloys based upon high mechanical strength and acceptable biosafety [3]. However, the development of chronic inflammation, bone resorption, and implant failure due to the stress-shielding effect (the mismatched elastic modulus between autologous bone and allogeneic materials) associated with long-term metal implants is a known disadvantage [4]. Implant-related infection represents another major and devastating complication that may lead to complex revision surgery or the complete removal of failed implants, and this is associated with increased morbidity and medical costs [5,6]. The infection rate of the primary placement of implants is less than 2%, whereas it increases considerably to 5–40% after the revision surgery [7].

It is known that the development of an immunodeficient fibrotic region at the implant and bone tissue interface is prone to bacteria colonization, leading to infection [8]. The onset of infection may occur within a time frame of six months or even years, and a high risk of infection remains as long as the implant is present. The mechanism of implant infection involves the contact and adherence of free bacteria relative to the implant’s surface, which form a bacterial biofilm that is difficult to remove [8]. The only way to avoid implant-related bacterial infection is “no implant”, whereby the degradation or resorption of the implant or its surface will deprive the bacteria of a surface upon which to grow. Biodegradable implants with antimicrobial efficacy might therefore be an ideal solution to prevent postoperative bone infections [9]. Promising candidate materials should possess a continuous antibacterial capability in order to deprive the adhesion of bacteria, with complete degradation after fulfilling their function [10].

Recently, biodegradable metals (BMs), including magnesium (Mg), iron (Fe), and zinc (Zn), have attracted considerable research interest for bone replacement, with the aim of eliminating the need for a second removal surgery and minimizing the implant-to-bone migration of stress shielding to maintaining bone regeneration [11]. Unlike Mg and Fe, Zn and its alloys exhibit moderate degradation behaviors and Young’s modulus, without hydrogen release as a corrosion by-product, and are considered the most promising biodegradable metals due to their ability to promote bone regeneration by stimulating and inhibiting osteoblast differentiation and osteoclasts, respectively [12,13,14,15]. Most interestingly, Zn^2+^ has anti-infectious properties and inhibits bacterial metabolic activity [16]. One of the most well-known antimicrobial mechanisms of Zn-based materials involves the entry of zinc oxide (ZnO) into the bacterial cytoplasm following physical damage to the cell membrane. The resulting formation of reactive oxygen species (ROS) contributes to bacterial cell death and the subsequent inhibition of biofilm formation [17,18,19]. Nevertheless, pure Zn metal has poor antibacterial activity with limited effects on bacterial infection, adhesion, and biofilm formation, which has to be further improved in order to meet clinical requirements [20].

To date, several strategies have been developed to treat and prevent the bacteria-associated infection of implant surfaces, such as the application of antibiotics or coatings with active antibacterial properties [20,21]. However, accumulated bacteria after infection appear to inhibit the penetration of antibiotics via the formation of bacterial biofilms as a barrier on the surface of implanted materials [22,23,24]. The efficiency of antibiotics is further hindered by bacterially acquired drug resistance, such as that observed in Staphylococcus aureus (*S. aureus*) against methicillin [25]. Hence, the most effective and safe strategies begin with the implant itself [26]. Different surface modification techniques have been developed in a series of preliminary studies to improve the antibacterial property of pure Zn. Gilani et al. demonstrated the enhanced antibacterial properties of pure Zn foil after surface anodic oxidation treatment due to the fabricated ZnO micro/nanohole and nanoparticles array [27]. Sun et al. constructed a homogeneous micro-/nano-ZnP structure on a pure Zn substrate via a surface coating technique and revealed excellent antibacterial properties against both *E. coli* and *S. aureus* [20]. However, the limitations of these investigations are the fabrication of non-permanent surface features and not addressing the effect of released antibacterial Zn ions [28]. Therefore, we prepared pure Zn with both nanostructures and high Zn ion release via surface acid etching in order to provide a novel one-step strategy for obtaining the more effective and permanent antibacterial properties of pure Zn [29]. In our previous study [30], we attempted to accelerate the biodegradation of Zn by treating it with a mixture of sulfuric (H_2_SO_4_) and hydrochloric (HCl) acid (SHA). While this approach showed promise, the high etching strength of SHA made it challenging to control etching times, which may have resulted in the inconsistent surface texturing of the treated Zn. Thus, it is necessary to explore a well-controlled method of Zn surface texturing that does not compromise the integrity of the bulk material and at the same time increases the surface area of the modified Zn surface. Furthermore, our previous work did not thoroughly analyze or investigate the effects of an increased surface area on bacterial functions.

Therefore, we hypothesize that a “moderate” or balanced acid can create homogeneous nanosurface texturing on biodegradable metal Zn surfaces in order to simultaneously increase the surface area and Zn ion release. The novelty of nanosurface texturing for enhancing the antibacterial effect of biodegradable metal zinc is the use of nitric acid etching to create small uniform surface structures that increase the surface area of zinc and thus its exposure to bacteria, resulting in more effective antibacterial action. This study represents a significant advancement in the field by investigating the previously unexplored relationship between zinc surface texturing and bacterial adhesion. In particular, we have developed a novel method for fabricating textured zinc surfaces that maximize the surface area while maintaining bulk material integrity, and we have thoroughly characterized the resulting bacterial adhesion and biofilm formation on these surfaces. Our findings not only shed new light on the mechanisms underlying bacterial interactions with zinc surfaces but also have important implications for the development of next-generation antimicrobial coatings and biomedical implants. Here, the effects of nitric-acid-etching on surface morphology, surface roughness, surface area, chemical, physical, and phase composition and the antibacterial properties of biodegradable Zn were investigated to further elucidate the potential for acid-etched Zn with respect to various biomedical applications.

## 2. Materials and Methods

### 2.1. Samples Preparation

Commercially extruded pure Zn (purity of 99.95%) rods were purchased from the Nilaco corporation, (Nilaco Bldg., 1-20-6 Ginza, Chuo-ku, Tokyo, Japan). The metal rods were wire cut into disks with a dimension of Φ10 mm × 1 mm with respect to thickness. Mechanically wet grinding was carried out using silicon carbide (SiC) papers (400–4000, Struers, Milton, Australia), subsequently ultrasonic cleaned in 100% ethanol for 15 min, and then dried at room temperature. The polished samples were then chemically treated with 5% and 15% nitric acid (HNO_3_) (Nitric acid, Sigma-Aldrich, Castle Hill, Australia) for 10 min at room temperature at 150 rpm of shaking, followed by washing twice with Milli-Q water, cleaning via ultrasonication for 10 min in 100% ethanol to remove residuals, and then air drying at room temperature. The acid-etched pure Zn samples were designated as 5% and 15% etched Zn. Before conducting bacterial experiments, both sides of the specimen disks were sterilized under UV light for 30 min.

### 2.2. Surface Characterization

The surface morphology and elemental composition of untreated (polished) and acid-etched samples were observed a using field-emission scanning electron microscope (FESEM; JEOL-JSM 7001F, Tokyo, Japan) equipped with energy-dispersive spectrometry (EDS) at an accelerating voltage of 5–7 Kv. Platinum sputter coating was performed using a 5 nm surface layer. The hydrophilicity of untreated and acid-etched Zn samples was assessed using a water contact angle (WCA) goniometer (OCA 15EC, Filderstadt, Germany). Briefly, the image of one droplet of deionized water was captured at 3 s after contact with the studied sample’s surface. Parallel measurements (3 specimens × 3 droplets) were conducted for each group. Atomic force microscopy (AFM) was conducted to assess 2D and 3D topographies and the parameters of surface roughness (Ra, Rq, and Rz), volume, and surface area variation of ground (untreated) and acid-etched Zn specimens. The phase composition of the nitric acid-etched Zn surface was characterized using Bruker D8 Advance X-ray diffractometer, which was applied at 40 kV and 40 m (Cu Ka radiation) within the 2θ range of 20–90° at a scan rate of 2°/min. Fourier transform infrared spectroscopy (FTIR, Nicolet iS20, Thermo Scientific, Waltham, MA, USA) was carried out to assess the degradation products on the surface within a wavelength range of 4000–400 cm^−1^. The surface’s chemical composition was further evaluated using a Bruker Advance D8 X-ray Diffractometer equipped with a LynxEye detector (Bruker, Billerica, MA, USA) at an operating voltage of 40 kV and 40 mA (Cu Ka radiation) within the 2θ range of 10–90° at a scanning rate of 0.02 s^−1^. An X-ray photoelectron spectroscopy (XPS) instrument (Kratos AXIS SUPRA PLUS, Manchester, UK) was then used to examine surface chemistry, with a monochromatic Al Kα X-ray source (hν = 1486.6 eV) operated at 10 mA of emission current and 12 kV of anode potential. XPS data were analyzed using Casa XPS software version 2.3.24 (Casa Software Ltd., Teignmouth, UK), and then the spectra were displayed after energy calibration with a reference peak of C1s at 284.5 eV.

### 2.3. Antibacterial Evaluation

#### 2.3.1. Biofilm Development

The antibacterial property of polished and nitric-acid-etched Zn specimens was evaluated using *Staphylococcus aureus* (*S. aureus*, ATCC 25923). Tryptic soy broth (TSB) was selected as the liquid medium. The *S. aureus* bacterial strain was frozen in stock overnight and cultivated in TSB media at 37 °C and 180 rpm; then, it was adjusted to a standard bacterial density of approximately 1 × 10^7^ colony-forming units (CFU)/mL using a spectrophotometer. The inoculum media were then prepared by mixing 1 mL of *S. aureus* suspension and culture media in a 1:10 final dilution ratio. The sterilized pure Ti, 5% etched Zn, and 15% etched Zn metal disks were placed in a 24-well plate with replicate samples in each group, and 1 mL of bacterial suspension was added to each well. Substrates were then incubated with bacteria in an anaerobic gas box at 37 °C for 24 and 72 h on a shaker.

#### 2.3.2. Bacterial Colony Number Counting

A plate colony number-counting assay was conducted to quantify the number of clonal bacterial colonies. After 24 and 72 h of culturing, the metal disks were rinsed three times in phosphate-buffered saline (PBS) to gently remove any non-adherent bacteria. The substrates were then placed in sterile Eppendorf tubes (5 mL) with 1 mL of PBS solution, and an ultrasonic (BIOSONIC UC300 Ultrasonic, Altstätten, Switzerland) bath was used to separate the adhered bacteria from the metal surface at 50 HZ for 15 min [31]. Thereafter, a gradational dilution was carried out on the detached bacterial suspension and plated on TSB agar plates incubated at 37 °C for 24 h. The number of bacterial colonies on the plates was counted by a colony counter (IUL DOT Colony Counter, ThermoFisher Scientific, Scoresby, Australia).

#### 2.3.3. Biofilm Metabolic Activity

Biofilm metabolic activity was determined using a tetrazolium salt, 2,3-bis (2-methoxy-4-nitro-5-sulfophenyl)-5-[(phenylamino) carbonyl]-2H-tetrazolium hydroxide (XTT; Sigma-Aldrich, Castle Hill, Australia) assay. The activated XTT solution was prepared by mixing 0.4 mM of menadione (Sigma-Aldrich) with PBS and XTT stock solution at a ratio of 79:20:1 (*v*/*v*/*v*). After the incubation of the material with the bacterial suspension for 24 and 72 h, the metal disks were mildly rinsed three times in PBS, followed by incubation with the well-activated XTT solution at 37 °C in the dark for 4 h. Thereafter, 100 μL of the co-cultured solution was transferred to a 96-well plate, and absorbance was then measured by a spectrophotometer (Tecan infinite 200 pro, Männedorf, Switzerland) at a wavelength of 492 nm.

#### 2.3.4. Crystal Violet Assay

The biofilm biomass of samples was qualitatively assessed by using a crystal violet assay after incubation with the bacterial suspension for 24 and 72 h. After gently rinsing three times in PBS, metal disks were placed into a fresh 24-well plate, and 500 μL of 0.1% crystal violet was added into each well to stain for 15 min at room temperature. The samples were then rinsed with 1 mL of PBS three times to remove excess crystal violet stain. Afterward, 1 mL of 10% acetic acid was added to each well for 15 min at room temperature to dissolve the unbound dye. In total, 100 μL of the solubilized solution was then transferred into a 96-well plate for absorbance detection at 570 nm using a spectrophotometer (Tecan infinite 200 pro).

#### 2.3.5. Confocal Microscopy

Biofilm formation and morphology were observed using a live/dead staining assay employing a Leica TCS SP5 scanning laser confocal microscope (Leica Microsystems, Mannheim, Germany). PBS resining was performed three times for the studied specimens. The untreated and treated Zn disks were then placed in a new 12-well plate and stained with 1 mL of SYTO 9 and propidium iodide (LIVE/DEAD BacLight Bacterial Viability Kit, Thermo Fisher, Bangkok, Thailand) at a 1:1000 dilution ratio in PBS and incubated in the dark at room temperature for 15 min. The confocal microscope was then used to image the stained biofilms. Live bacteria were stained using gree-fluorescent SYTO 9 and observed at the Exc/Em wavelengths of 488/526 nm; the dead bacteria were stained with red-fluorescent PI and observed at the Exc/Em wavelengths of 493/636 nm at 10×. The biofilm’s thickness was also examined using CLSM.

#### 2.3.6. Surface Morphology Observation by SEM

Bacterial morphology was also observed using scanning electron microscopy (SEM). As mentioned above, the sample metal disks were placed in a new 12-well plate after PBS rinsing 3 times and were fixed with 4% paraformaldehyde (PFA) for 20 min. The fixed substrates were then washed with PBS three times to remove the remaining PFA. A dehydration process using an ethanol series (50%, 70%, 90%, and 100% 2 times for each concentration) was carried out to prepare SEM samples. The samples were then air dried and sputter coated with platinum and observed using SEM (FESEM; JEOL-JSM 7001F, Tokyo, Japan).

### 2.4. Statistical Analysis

Data were expressed as mean ± standard deviation and statistically analyzed within GraphPad Prism^®^ (Accessed on 1 October 2022, version 9.0.0 for Windows, GraphPad Software, La Jolla, CA, USA, www.graphpad.com). One-way analysis of variance (ANOVA) was conducted between different groups, followed by post hoc Turkey’s multiple comparison tests. The significance of statistical data was considered at a *p*-value of <0.05 (*) (**, *p* < 0.01; ***, *p* < 0.001; ****, *p* < 0.0001).

## 3. Results

### 3.1. Surface Characterization

Figure 1a–c present the surface morphology and microstructure of untreated 5%, and 15% nitric-acid-etched Zn samples, respectively. The surface roughness of biodegradable pure Zn can significantly increase using acid etching, as previously reported [30]. SEM images confirmed smooth surface morphology for untreated Zn. In contrast, numerous non-uniformly distributed protuberances formed on the 5% etched Zn surface, with increased spacing between neighboring needle-/leaf-like structures observed at higher concentrations (15%) of nitric acid (Figure 1b,c). More interestingly, the needle-/leaf-like nano-crystallites on the 15% etched Zn surface exhibited dimensions of 1106 ± 93 nm in length and 135 ± 14 nm in width, which are significantly larger than that of 5% etched Zn with nano-crystallites measuring 229 ± 28 nm in length and 20 ± 4 nm in width. This is attributed to the fast dissolution of ZnO due to the hydrodynamic convection of H^+^ that is driven by water and the increased nitric acid accompanied by high H_2_O contents [32]. As shown in the EDS mapping images of Figure 1d,e, the surface elemental composition of 15% nitric acid etching is mainly composed of Zn with an average atomic % of 71.44 ± 7.79, indicating a significant dissolution of the surface ZnO layer due to the relatively strong chemical etching, resulting in the increased exposure of raw materials [33]. In contrast, the 5% etched Zn surface is rich in O, with an average atomic % of 43.20 ± 3.25, suggesting the presence of a more significant build up of the ZnO layer formed during the early stage reaction with relatively weaker nitric acid [33]. The surface wettability of untreated and treated Zn was evaluated, and the results are displayed in the inset of the SEM images of Figure 1a–c. The untreated Zn control showed a hydrophobic surface with a WCA of 82.90 ± 1 degree (Figure 1a). However, after acid etching, the etched Zn surface exhibited significantly higher wettability (reducing the WCA), where the WCA for 15% and 5% etching was 51.10 ± 8 and 64.90 ± 2 degrees, respectively (as shown in Figure 1b,c). Such effects of the surface topography on the substrate’s wettability were established by Wenel et al. according to the following equation:r = (cos θw)/(cos θe) 
where r is the surface roughness factor, and θw and θe are the actual contact angle and geometric contact angle, respectively. The surface roughness of the acid-etched Zn significantly increased, hence enhancing the cos θw value and decreasing the actual contact angle (θw) of the interface, representing increased wettability [34]. The etched Zn surfaces exhibiting porous nano-tomography structure can induce a capillary effect that is potentially responsible for the decreased water contact angle and enhanced hydrophilicity [35]. Interestingly, the 15% etched Zn surface exhibited higher hydrophilicity than that of the 5% etched Zn surface, suggesting the greater crystallite size and porosity with respect to surface topography and nanostructures, which are clearly evident in the current SEM images [36]. Importantly, a hydrophilic-based implant surface promotes early protein adhesion and cell attachment, accelerating the wound-healing response after implantation [37].

Further exploration of the effect of nitric acid etching on the surface roughness profile of Zn samples included 3D AFM images, link scan measurements of the 3D images, surface roughness quantitative analyses (Rz, Rq, and Ra), and volume and surface area variations (Figure 2).

It can be observed in the 3D images that untreated Zn exhibited a smooth surface with parallel scratches originating from the mechanical grinding process (Figure 2a,d). However, after the acid treatment, both 5% and 15% nitric-acid-etched Zn (Figure 2b,c,e,f) showed significantly increased surface roughness with intensely distributed needle-/leaf-like structures, in agreement with the obtained SEM results. Surface roughness quantification (Figure 2g–i) demonstrated that 15% etched Zn possessed the highest root-mean-squared roughness Rq at 28.68 ± 1.34 nm and an arithmetic average roughness Ra at 21.79 ± 1.62 nm compared with untreated Zn (Rq at 3.44 ± 0.52 nm and Ra at 2.58 ± 0.48 nm) and 5% etched Zn (Rq at 25.72 ± 2.90 nm and Ra at 19.35 ± 2.58 nm), resulting in larger nano-crystal structures. The surface-to-volume ratio of 15% etched Zn was lower than 5% etched Zn (Figure 2h) due to the reduced size of crystal structures, and the ratio is inversely proportional to the characteristic length of the surface particle [38]. The average peak-to-valley separation distance (Rz) can be ranked as untreated Zn > 5% etched Zn > 15% etched Zn, which is consistent with Ra and Rq data (Figure 2i). The higher surface roughness after relatively strong acid etching is most likely contributed to by the rapid water-selective dissolution of ZnO in the more exposed surface and the creation of a large gap between the neighboring dendritic-like structure, thus leading to increased surface roughness with sharp and thin needle-/leaf-like structures [33,39]. Additionally, the porosity of the produced nanostructure also increases with the concentrated acid, thereby increasing surface roughness [40].

Figure 3a, Table 1 and Appendix A show the XRD profiles and their data analyses of untreated and acid-etched Zn samples, respectively. The XRD profile demonstrates that the characteristic peaks display a similar trend over a 2θ range of 40–60° for untreated Zn metal (PDF Card—01-085-5877). The existence of the diffraction peaks below 35 degrees in 2 theta was intended to be the (002) plane of ZnO, suggesting surface oxidation during the sample preparation process [41]. Interestingly, 15% etched Zn displayed the sharpest diffraction peaks at 101 and 102 (ZnO) compared with the broadened diffraction peaks of untreated Zn, suggesting a transformation of the crystal structure from the amorphous phase to concentrated crystallization due to the dissolution of the surface resulting from the relatively strong acid [33]. In contrast, the gradually decreased diffraction peaks at 103, 004, and 112 along with the increased acid concentration indicate a change in the lattice constant [42]. The average crystal size of untreated and acid-etched Zn was determined by calculating the full width at the half maximum (FWHM) using Scherrer’s equation [43], as displayed in Table 1. The nitric-acid-etched Zn at 5% concentration presents a transition of crystal size from microscale to nanoscale (it particularly decreased from 202.60 nm to 50.66 nm at 2-θ of 39.10°, in line with SEM results). With nitric acid concentrations increased from 5% to 15%, the XRD diffraction peaks are sharper and disappear, contributing to the decreased FWHM compared with the untreated Zn [33].

X-ray photoelectron spectroscopy (XPS) was conducted to further characterize the chemical composition of the sub-surface and the elemental binding states of untreated and nitric-acid-etched Zn samples (Figure 3b). It can be observed that the untreated and etched Zn present consistent trends with typical Zn2p, O1s, and C 1s spectra (Figure 3b), indicating the non-additional inputting of impurities or other materials from the acid and only altering the phase and chemical structure of the surface after treatment [30]. As shown in Table 2, the slight reduction of O composition from 15% etched Zn in comparison to 5% etched Zn is attributed to the dissolution of the superficial ZnO structure with increased acid concentrations that aligned with the XRD analysis [44]. It can be noted that a difference in oxygen concentrations was observed in the EDS and XPS data. This can be attributed to the different analysis depths used in these techniques [45]. EDS analysis can provide micrometer-scaled insights into the material, while XPS is a surface-sensitive technique that only analyzes the top 1–10 nanometers of the material [46]. Hence, variation was observed in the surface’s composition when comparing EDS and XPS data. A deeper exploration of the effects of the nanometric crystal size of implanted materials contributes to establishing a better understanding of its bioactivity, biocompatibility, biodegradation, osteoinductivity, and mechanical properties for further clinical translation [47].

The surface chemical composition after nitric acid etching was further confirmed using the FTIR spectra, as illustrated in Figure 4. FTIR untreated and acid-treated Zn displayed similar patterns, and the corresponding bands indicated O-H and ZnO bonds. The peaks located at 3300 cm^−1^ and 1500 cm^−1^ are characteristic of the broad bond of O-H stretching vibrations [48]. The narrow peak at 560 cm^−1^ corresponds to ZnO, implying the presence of ZnO after acid treatment, and the intensity decreased with the increased acid concentration [49].

To summarize, nitric acid etching maintains the surface phase and chemical composition of biodegradable pure Zn, while effectively endowing them with permanent functions driven by various constructed novel nanostructures [28]. The micro-/nanostructure dimensions of implanted materials have been reported as exhibiting strong antibacterial efficiency, indicating that the nitric-acid-etched Zn surface with specific nanostructure may impact bacterial formation [50,51].

### 3.2. Antibiofilm Properties

The antibacterial activity performances of the studied substrates were evaluated using XTT, crystal violet, and live/dead staining assay after 1 and 3 days of incubation (Figure 5a–d). There was substantially less biofilm metabolic activity on nitric-acid-etched Zn specimens, especially 15% etched Zn compared to the pure Ti and untreated Zn, showing continually decreased anti-biofilm activity and mass properties (Figure 5a,b). This contributed to the increased surface roughness and surface area with high Zn contact with bacteria, inhibiting metabolic activity. As shown in Figure 5c,d, *S. aureus* exposed to the surface of pure Ti substrates remained alive over the entire exposure period, while there was a significant decline in live bacteria on the surface of acid-etched Zn substrates after 24 h of culture, and viability decreased extremely after 3 days of exposure, suggesting the great antibacterial ability of the treated Zn substrates. The bacterial morphology and antibacterial efficacy of the studied substrates were further examined using direct visualization via SEM and a bacterial monoclonal number-forming assay (Figure 6a–c). As observed in SEM images in Figure 6a, multi-layered *S. aureus* with spherical and intact cell membranes stack on the surface of Ti samples. However, the adhered bacterial number of nitric-acid-etched Zn decreased significantly, and the cell membranes changed to become corrugated and distorted or even completely damaged due to the higher concentration of Zn ion release than that of untreated Zn, indicating enhanced antibacterial performances with respect to preventing biofilm adhesion and development [52]. Additionally, after detaching bacteria from the surface of metal disks followed by serial dilution and incubation in the agar plate for 24 h and 72 h, a large number of formed bacterial colonies were observed on the Ti and untreated Zn samples, suggesting no effects and slightly inhibitive effects on *S. aureus* formation (Figure 6b), respectively. In contrast, an approximate 15- and 40-fold reduction in colony numbers was observed in the 5% and 15% acid-etched Zn substrates, respectively, consistent with results obtained using SEM and quantitative XTT and CV assays (Figure 5a,b).

Zn has been widely applied as an antibacterial material or substrate, such as ZnO films, coated Zn-incorporated nanotube arrays, and cross-linked nanocomposite scaffolds [6,53]. The inherent antimicrobial property of Zn ions is driven by interacting with the bacterial surface and disrupting the membrane’s charge balance in order to encourage cell anti-adhesion, deformation, and bacteriolysis [54]. In the present study, nitric-acid-etching-mediated Zn ion accumulation negatively impacted bacterial metabolic activity and survival (Figure 6a). Dai reported a Zn-0.8Li-0.5Ag alloy exhibiting rare biofilm formation and bacteria formation due to the accumulated Zn ion release, preventing the colonization of the material’s surface [26]. The adhesion and formation of bacteria were also significantly affected by the surface roughness of the substrate material [55]. It was previously reported that increased surface roughness from combined micro- and nanostructures can impede bacterial division and attachment due to the forced deformation stress on the wall and inner membrane of the bacteria [56,57]. Su and colleagues have demonstrated that a ZnP-coated Zn substrate with micro-/nano-scaled structures presents a rough and large surface area for inhibiting bacterial adhesion and can significantly improve antibacterial performance, which is consistent with our study [20]. Overall, nitric-acid-etching can be considered a promising method for improving the antibacterial performance of Zn-based biodegradable implants and for further serving other medical applications due to its easy processing and stable nanoscale structures.

## 4. Conclusions

Currently, the bacteria-related infection of implanted materials occurs commonly when inserted into the body, leading to a high risk of complication and potential implant failure and resulting in pain and health care costs for revision surgery. Biodegradable Zn as a promising material has attracted substantial research interest with respect to the solution to current clinical challenges due to its self-antibacterial properties, whereas its anti-infection efficiency is still far from clinical requirements. Hence, in this study, we have performed a straightforward nitric acid etching (5% and 15%) modification on biodegradable pure Zn to simultaneously enhance its surface roughness and ion release and subsequently demonstrated desirable and effective antibacterial properties resulting from the unique surface nanostructures and the increased presence of Zn^2+^ ions. Moreover, the 15% nitric-acid-etched Zn displayed a strong antibacterial property by inhibiting bacterial adhesion and biofilm formation. Overall, the nitric acid etching technique is an effective method for improving the antibacterial properties of biodegradable zinc-based materials, making them more suitable for use in biomedical applications.

## Figures and Tables

**Figure 1 nanomaterials-13-02022-f001:**
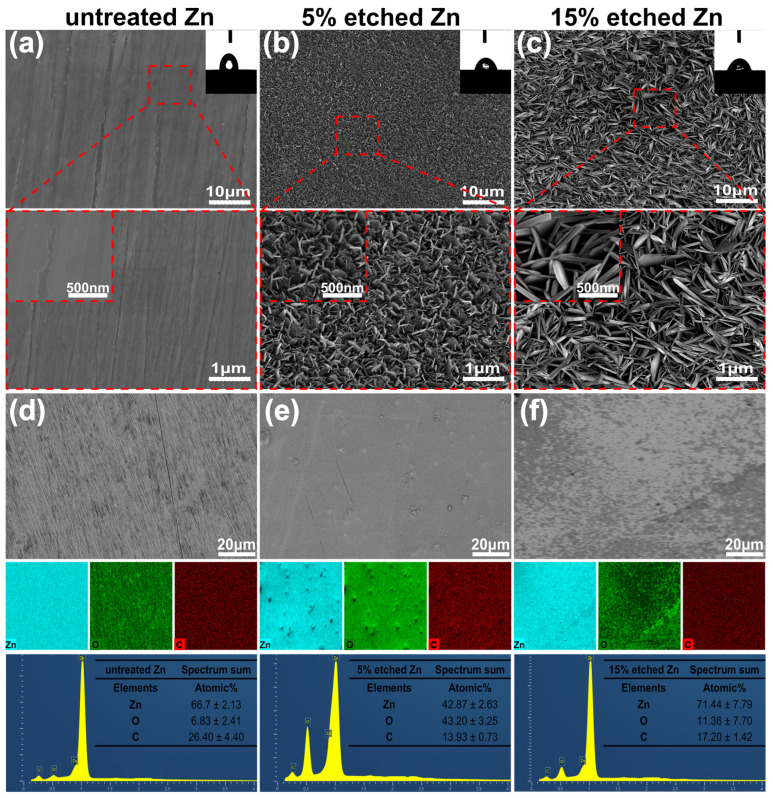
SEM surface morphology, EDS mapping, and elemental fraction. (**a**–**c**) Microstructures of the untreated Zn, 5% etched Zn, and 15% etched Zn, respectively. (**d**–**f**) Corresponding elemental composition (EDS) and the atomic element percentage of Zn, O, and C in the selected region of Figure (**d**–**f**). The 2D images, the average data of the water contact angle’s analysis, and the magnified SEM images of untreated and acid-etched Zn samples are shown in the inset of Figure (**a**–**c**).

**Figure 2 nanomaterials-13-02022-f002:**
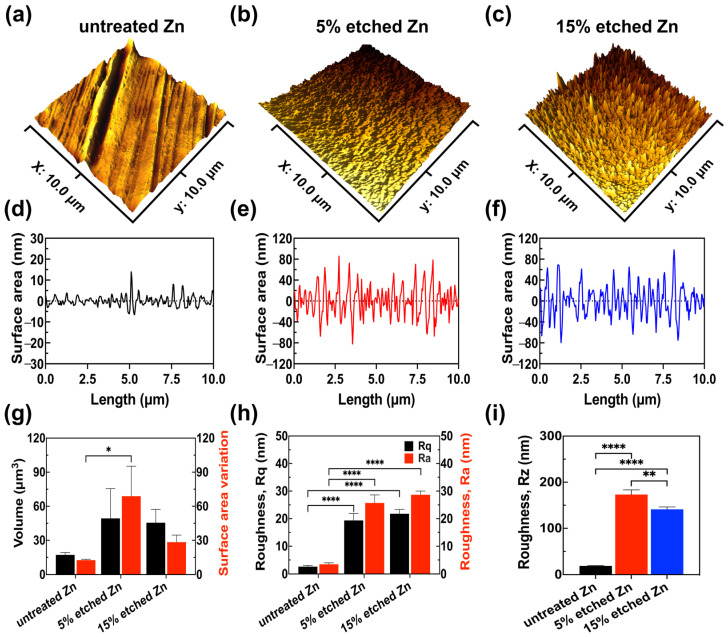
AFM results reveal the surface roughness profile of untreated, 5% etched, and 15% etched Zn. (**a**–**c**) Three-dimensional surface topography. (**d**–**f**) Roughness from the line scan in 3D images. (**g**) Root-mean-square roughness along the measuring length (Rq) and the average arithmetical value of all absolute distances of the roughness profile from the center line within the measuring length (Ra). (**h**) Volume and surface area variation. (**i**) The average value of the absolute values of the heights of the five highest profile peaks and the depths of the five deepest alleys within the evaluation length (Rz). The significance of statistical data was considered at a *p*-value of <0.05 (*) (**, *p* < 0.01; ****, *p* < 0.0001).

**Figure 3 nanomaterials-13-02022-f003:**
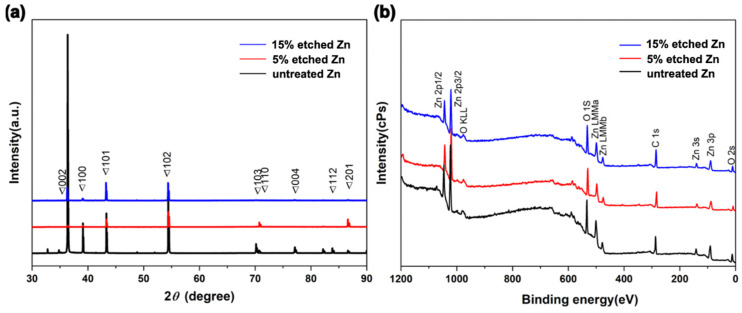
Compositional surface analysis of untreated and nitric-acid-etched Zn specimens. (**a**) X-ray diffraction spectra. (**b**) Survey XPS spectra.

**Figure 4 nanomaterials-13-02022-f004:**
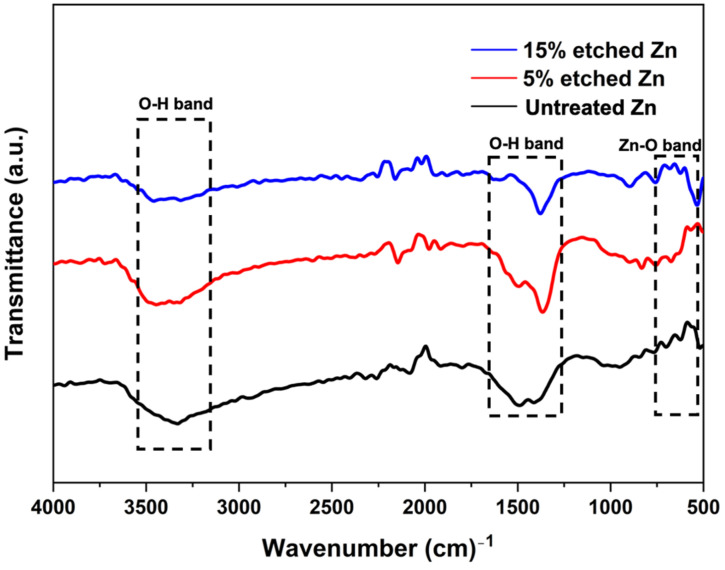
Fourier transform infrared spectroscopy (FTIR) spectra of untreated and nitric-acid-treated Zn samples at 5 and 15%.

**Figure 5 nanomaterials-13-02022-f005:**
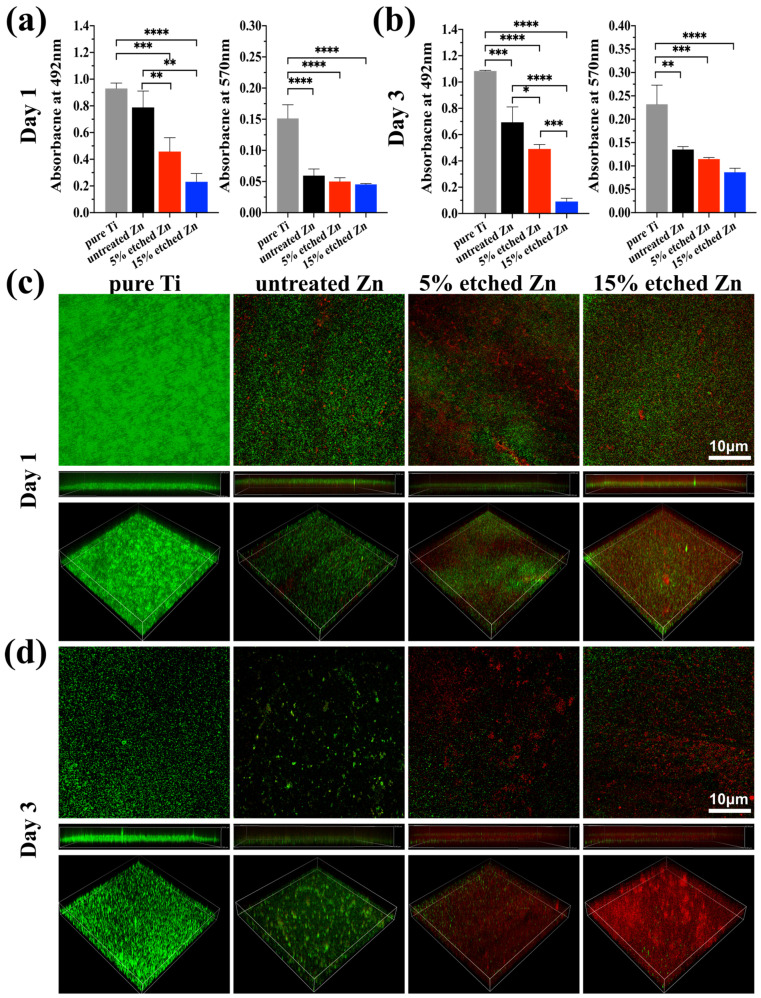
Assessment of bacterial performances on untreated and nitric-acid-etched Zn substrates with a commercial pure Ti disc as the control. (**a**,**b**) XTT and crystal violet assay on 1 and 3 days, respectively, assessing biofilm metabolic activity and the entire biomass. (**c**,**d**) Confocal fluorescent images of bare Ti, Zn, and treated Zn specimens with live bacterial (green) and dead (red) staining, respectively, after 1 (**c**) and 3 (**d**) days of bacterial culture. The significance of statistical data was considered at a *p*-value of <0.05 (*) (**, *p* < 0.01; ***, *p* < 0.001; ****, *p* < 0.0001).

**Figure 6 nanomaterials-13-02022-f006:**
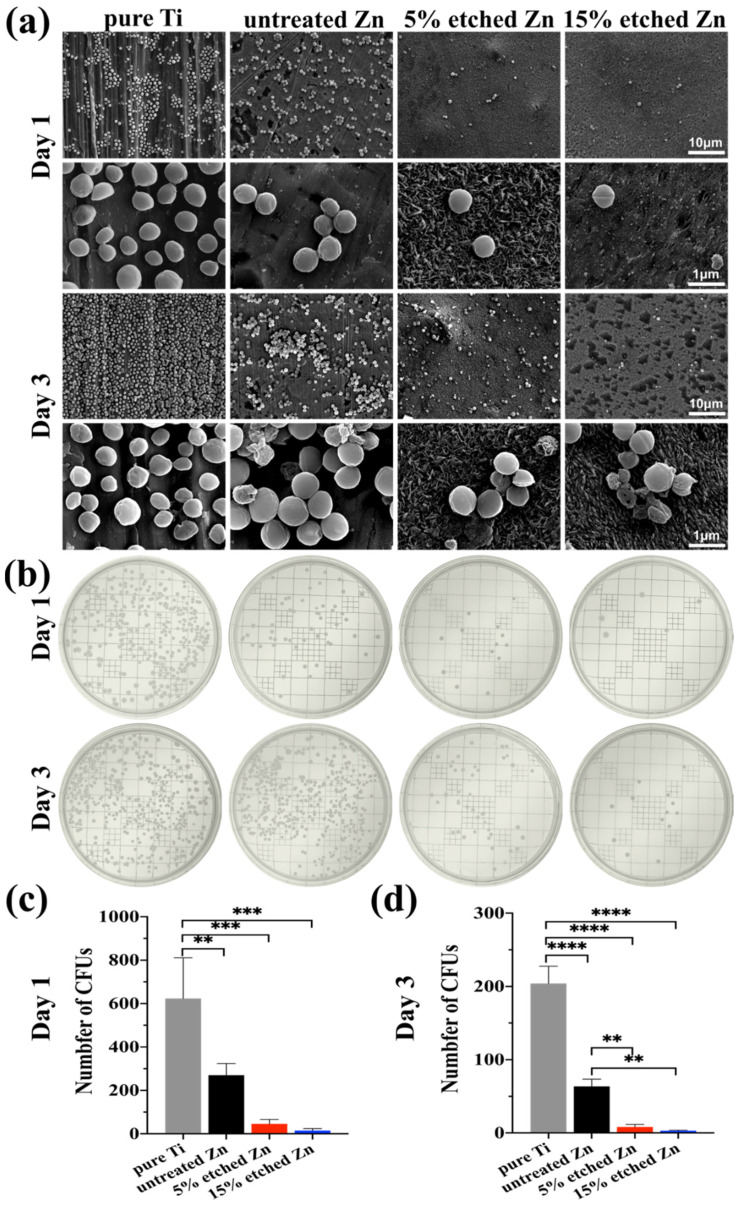
*S. aureus* biofilm morphology and colony formation after incubation for 1 and 3 days with the corresponding bacterial suspension detached from the surface of substrates. (**a**) Representative SEM images of bacteria grown in different substrates after 1 and 3 days, with pure Ti as the control. (**b**) Typical images of the *S. aureus* colony formation assay after 1 and 3 days of growth on TSB agar plates. (**c**,**d**) Quantitative colony formation assay. The significance of statistical data was considered at a *p*-value of <0.05 (**, *p* < 0.01; ***, *p* < 0.001; ****, *p* < 0.0001).

**Table 1 nanomaterials-13-02022-t001:** XRD crystallite size analysis.

	K	Λ (Å)	Peak Position (°)	FWHM (RAD)	L (nm)	Average (nm)
untreated Zn	0.94	1.54	36.39	0.13	68.51	151.63
	0.94	1.54	39.10	0.04	202.60	151.63
	0.94	1.54	43.32	0.06	151.93	151.63
5% Zn	0.94	1.54	36.33	0.23	38.21	76.95
	0.94	1.54	39.09	0.17	50.66	76.95
	0.94	1.54	43.25	0.06	141.99	76.95
15% Zn	0.94	1.54	36.42	0.15	59.20	103.51
	0.94	1.54	39.13	0.07	120.56	103.51
	0.94	1.54	43.32	0.07	130.77	103.51

**Table 2 nanomaterials-13-02022-t002:** The chemical composition concentration of untreated (polished) and nitric-acid-etched Zn samples (At%) was obtained from XPS profiles (Figure 3b).

	O1s	C1s	Zn2p
untreated Zn	36.71	55.17	8.11
5% etched Zn	35.36	57.89	6.75
15% etched Zn	34.23	59.86	5.91

## Data Availability

The data presented in this study are available upon request from the first author.

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
