# Peer review of "Nanosurface Texturing for Enhancing the Antibacterial Effect of Biodegradable Metal Zinc: Surface Modifications"

_nanomaterials, 2023, doi:10.3390/nano13132022_

Round 1

Reviewer 1 Report

The untreated sample has some unidentified diffraction lines, below 35 2theta degrees. Could be the precursors? Please, verify!

The rod-shaped nano-crystallites seems to be an inappropriate definition for the particles morphology. From SEM images it is difficult to identify rod-shaped crystals. I understand by rod-shape a morphology with an Lxl section and a length to section high aspect ratio. In the images from the manuscript, a scales-like shape can be distinguished. Furthermore, for a rod-shape nano-crystallite ZnO, the (002) plane should be the most prominent diffraction line in the pattern.

Please, use the same decimal system all over the manuscript. Two decimals will be representative.

The presented biological characterizations are not enough to conclude that the materials is fit for medical applications. Further tests, as biodegradation, mechanical and citotoxicity tests are necessary to a complete biological evaluation.

Author Response

Response to Reviewer 1 Comments

Point 1: The untreated sample has some unidentified diffraction lines, below 35 2theta degrees. Could be the precursors? Please, verify!

Response 1: We appreciate the reviewer’s comments. We have re-evaluated the original data for the untreated samples and confirmed that they are composed of pure Zn. Please see the attached section of PDF card (Figure 1) for further information. The existence of the diffraction peaks below 35 2θ degrees was intended to be the (002) plane of ZnO, suggesting surface oxidation during the sample preparation process 1. It is crucial to emphasize that Zn is a highly reactive metal capable of undergoing a reaction with oxygen present in the air. This reaction results in the formation of a delicate layer of zinc oxide on the surface, a process that occurs even under typical atmospheric conditions 2. The untreated Zn specimens were prepared via mechanical wet-grinding using silicon carbide (SiC) papers (1200-4000), a common method employed for preparing biodegradable metals and alloys prior to in vitro and in vivo study 3. The fresh Zn surface created by the grinding process is even more easily oxidized during subsequent analysis. However, the weak intensity of the detected diffraction peak suggests a minimal crystal formation in that specific orientation, contributing a limited effect on the overall crystal structure. Moreover, in the FTIR spectrum, there is no strong Zn-O absorption band of untreated Zn in the 400-500 cm-1 range to fit the stretching mode of the Zn-O bonds in the ZnO lattice. We highlight that the term ‘precursor’ is used to indicate the initial material that undergoes transformations to create a specific crystal structure. Despite this, our XRD pattern after acid etching did not exhibit any new crystal structure. Instead, only the changes in the Zn crystal structure from the strong reaction between Zn metal and nitric acid were observed. We have revised the manuscript accordingly, on lines 310 to 312 of page 8.

Figure 1 Section of pure Zn XRD card.

Point 2: The rod-shaped nano-crystallites seems to be an inappropriate definition for the particles morphology. From SEM images it is difficult to identify rod-shaped crystals. I understand by rod-shape a morphology with an Lxl section and a length to section high aspect ratio. In the images from the manuscript, a scales-like shape can be distinguished. Furthermore, for a rod-shape nano-crystallite ZnO, the (002) plane should be the most prominent diffraction line in the pattern.

Response 2: We thank the reviewer for the observation, and agree with the suggestion. Instead of a rod-shaped structure, the morphology of the nitric-acid etched Zn surface is referred to as exhibiting a needle-like and leaf-like structure. The manuscript has been modified accordingly.

Point 3: Please, use the same decimal system all over the manuscript. Two decimals will be representative.

Response 3: We thank the reviewer for this suggestion. We have revised the manuscript to ensure consistent use of the decimal system throughout, with all data now presented to two decimal places.

Point 4: The presented biological characterizations are not enough to conclude that the materials are fit for medical applications. Further tests, as biodegradation, mechanical and citotoxicity tests are necessary to a complete biological evaluation.

Response 4: We appreciate the reviewer’s feedback and agree that comprehensive biological characterizations are necessary to advance use of etched Zn for medical applications. The presented study focused primarily on enhancing the anti-infection properties of biodegradable Zn via nitric acid etching, in response to the lasting challenges in tissue therapy and particularly the need for effective materials that can prevent infection. As shown in Figure 6 of our manuscript, we believe that our data effectively demonstrate the promising anti-bacterial capabilities of the acid-etched zinc. While we understand the necessity of conducting a complete biological assessment including biodegradation, mechanical, and cytotoxicity evaluation, these additional analyses were not within the scope of this paper. However, they will certainly be included in our future work. We hope that our focus of enhancing the anti-infection properties of nanostructured Zn will contribute deeper understanding of its potential in the field of nanostructured biomaterials for tissue repair and anti-infection.

REFERENCE

  1. Bindu, P.; Thomas, S., Estimation of lattice strain in ZnO nanoparticles: X-ray peak profile analysis. Journal of Theoretical and Applied Physics 2014, 8, 123-134.
  2. Hammer, G.; Shemenski, R., The oxidation of zinc in air studied by XPS and AES. Journal of Vacuum Science & Technology A: Vacuum, Surfaces, and Films 1983, 1 (2), 1026-1028.
  3. Yang, H.; Jia, B.;  Zhang, Z.;  Qu, X.;  Li, G.;  Lin, W.;  Zhu, D.;  Dai, K.; Zheng, Y., Alloying design of biodegradable zinc as promising bone implants for load-bearing applications. Nature communications 2020, 11 (1), 401.

Reviewer 2 Report

The authors report an innovative one-step surface modification method, implemented by nitric acid treatment, prone to accelerate the Zn ions release via increasing surface roughness, thereby endowing an effective antimicrobial property and biofilm formation inhibition and have demonstrated the effect of the functionalized surface against Staphylococci aureus biofilm. The reported technique proved to be an effective way to produce unique morphological surface structures and enhance the antibacterial properties of biodegradable Zn-based materials.

In my opinion, several points need to be improved/clarified.

1. In Section 2, subsection 2.3.2 the authors use ultrasound to separate the adhered bacteria from the metal surface. Doesn’t US cause damage to the treated cells?

2. At which accelerating voltage were the EDS maps acquired? Depending on the accelerating voltage value, the penetration depth of the electron beam varies. Is it possible, that the acid, used for etching, has left traces on the processed sample (due to its complex morphology), but was not detected due to the high accelerating voltage?

3. The insets in Fig. 1 in the second row with 500-nm scale bar are hard to distinguish from the full-size images. Perhaps it would be better to contour their edges.

4. The authors should also perform a more detailed analysis of the FT-IR spectra. The band at ~1400 becomes more prominent after the etching; the band at ~1416 cm-1 may result from the NO3¯; at 1323 cm-1 - from NH molecular vibrations, if there is any nitric acid residue left.

Author Response

Response to Reviewer 2 Comments

Point 1: In Section 2, subsection 2.3.2 the authors use ultrasound to separate the adhered bacteria from the metal surface. Doesn’t US cause damage to the treated cells?

Response 1:  We appreciate the reviewer’s viewpoint. In Section 2.3.2, we describe using ultrasonication to detach bacteria adhered to the metal disc into 1ml of PBS. This is a well-established protocol from the National Standard of China GB/T 4789.2 1-4 that has been utilized in numerous studies determining the bacteriostatic efficacy on metal disc-based materials since 2001. The frequency and intensity of the ultrasound are key in facilitating bacteria detachment. The reference is now cited in the revised manuscript: line 177 of page 4.

Point 2: At which accelerating voltage were the EDS maps acquired? Depending on the accelerating voltage value, the penetration depth of the electron beam varies. Is it possible, that the acid, used for etching, has left traces on the processed sample (due to its complex morphology), but was not detected due to the high accelerating voltage?

Response 2: We thank the reviewer for the valuable comments. We used a relatively low accelerating voltage (5-7kv) for the EDS mapping in our study. Based on our experience with researching the surface modification of zinc, it is important to allow the electron beam to interact with the near-surface regions of the sample. On the other hand, lower accelerating voltages can offer better spatial resolution due to the smaller interaction volume of the electrons with the sample, allowing for more precise mapping of the element distribution 5. Regarding the reviewer's question about potential traces of the etching acid, we confirm that no acid residues were detected for all the etched samples from the EDS mapping data, as no nitrogen elements were observed. This can be attributed to the effective removal of residuals during the cleaning steps. The value of accelerating voltage was added in the revised manuscript: line 137-138 of page 4.

Point 3: The insets in Fig. 1 in the second row with 500-nm scale bar are hard to distinguish from the full-size images. Perhaps it would be better to contour their edges.

Response 3: We thank the reviewer for the valuable suggestion. The insets provided in Figure 1 (second row) have been modified with enhanced edge contours for improve clarity. Please see the modified Figure 1, Page 7.

Point 4: The authors should also perform a more detailed analysis of the FT-IR spectra. The band at ~1400 becomes more prominent after the etching; the band at ~1416 cm-1 may result from the NO3¯; at 1323 cm-1 - from NH molecular vibrations, if there is any nitric acid residue left.

Response 4: We appreciate the reviewer’s comments and suggestions. We agree that the band observed at around 1400 cm-1 in the FTIR spectra becomes more prominent after the etching. However, we believe it can be attributed to the O-H bending vibrations due to the specific bonding on the etched Zn surface. Regarding the potential presence of NO3- or NH groups, we determined other measures to clarify this point. If there is potential presence of NO3- or NH, we would observe the N1s peak located around 398-400 eV from XPS data due to the strong interaction between nitrogen and oxygen in these compounds. Moreover, we may also observe the Zinc Nitrate Hexahydrate (Zn (NO3)2.6H2O) diffraction peak at approximately 2θ values of 11.6°, 23.5°, 34.8°, 36.8°, and 47.5° according to Joint Committee on Powder Diffraction Standards (JCPDS) data card number 01-076-0700. However, the presented XRD and XPS data are not exhibiting these characteristics, demonstrating the complete elimination of nitric acid at the etched surfaces.

REFERENCE

  1. Zhao, L.; Wang, H.;  Huo, K.;  Cui, L.;  Zhang, W.;  Ni, H.;  Zhang, Y.;  Wu, Z.; Chu, P. K., Antibacterial nano-structured titania coating incorporated with silver nanoparticles. Biomaterials 2011, 32 (24), 5706-5716.
  2. Ye, J.; Li, B.;  Li, M.;  Zheng, Y.;  Wu, S.; Han, Y., Formation of a ZnO nanorods-patterned coating with strong bactericidal capability and quantitative evaluation of the contribution of nanorods-derived puncture and ROS-derived killing. Bioactive Materials 2022, 11, 181-191.
  3. Qu, X.; Yang, H.;  Jia, B.;  Wang, M.;  Yue, B.;  Zheng, Y.; Dai, K., Zinc alloy-based bone internal fixation screw with antibacterial and anti-osteolytic properties. Bioactive Materials 2021, 6 (12), 4607-4624.
  4. Qu, X.; Yang, H.;  Jia, B.;  Yu, Z.;  Zheng, Y.; Dai, K., Biodegradable Zn–Cu alloys show antibacterial activity against MRSA bone infection by inhibiting pathogen adhesion and biofilm formation. Acta Biomaterialia 2020, 117, 400-417.
  5. Small, J., The analysis of particles at low accelerating voltages (≤ 10 kV) with energy dispersive X-ray spectroscopy (EDS). Journal of research of the National Institute of Standards and Technology 2002, 107 (6), 555.

Reviewer 3 Report

I carefully review the whole article entitled “Nano Surface Texturing for Enhancing Antibacterial Effect of Biodegradable Metal Zinc: Surface Modifications”. In this work, the authors have prepared a nano surface texturing on biodegradable metal Zn via acid etching. As a result, the antibacterial effect were enhanced by the large surface area and elevated ion concentration. The content is novel and should be of interest to the reader of “nanomaterials”. Some queries and suggestions were showed as follows:

1. From my own perspective, ZnO is a typical hydrophobic substance, which is due to the presence of hydroxyl groups on its surface. However, 5% etched surface possess more ZnO but showed lower WCA than those of others surface. Please explain.

2. XPS revealed that the intensity of O1s spectrum derived from all specimens was similar. But the O concentration determined by EDS were very different. Please explain.

3. There is no peak originated from ZnO were detected in etched surface by XRD, but some peaks belong to ZnO were observed in the XRD profile of untreated surface. Please explain. 

4. There are some format errors. For instance, line 128 “HNO3” should be “HNO3”; The unit of WCA should be “degree”; line 237 “H+”; line 238 “H2O”.

Author Response

Response to Reviewer 3 Comments

Point 1: From my own perspective, ZnO is a typical hydrophobic substance, which is due to the presence of hydroxyl groups on its surface. However, 5% etched surface possess more ZnO but showed lower WCA than those of others surface. Please explain.

Response 1: We thank the reviewer for their insight. The nature of hydrophobic or hydrophilic properties of ZnO depends on the physical and chemical properties of ZnO/solution interfacial interactions 1. However, in general, zinc oxide is intrinsically hydrophilic due to the presence of hydroxyl groups on its surface. This aligns well with the findings of Pesika et al who reported that the surface of a ZnO single-crystal is relatively hydrophilic 2. In a solution environment, the growth unit of a ZnO crystal is the complex formed by the connection of cation with OH ions 3. The presence of relatively increased ZnO on the 5% etched surface increases the hydrophilicity as demonstrated by a reduced water contact angle (WCA).

Point 2: XPS revealed that the intensity of O1s spectrum derived from all specimens was similar. But the O concentration determined by EDS were very different. Please explain.

Response 2: We are very grateful for the reviewer’s observation. XPS and EDS are both analytical techniques commonly used to determine the elemental composition of the sample, although they have different working patterns 4. XPS is a surface-sensitive technique, generating composition information from the top 1-10 nanometers only of the material 5. We have obtained a similar O1s spectrum from XPS in all samples which might reflect the formation of a thin oxide layer, as zinc is highly reactive with the oxygen in air even during sample preparation. On the other hand, EDS provides insight to a depth of micrometers, covering a much larger volume compared to XPS 6. As such, we observed a different surface composition profile from EDS and XPS data. The difference in the concentration of oxygen from EDS and XPS is discussed in the revised manuscript: lines 333 to 338 of page 9.

Point 3: There is no peak originated from ZnO were detected in etched surface by XRD, but some peaks belong to ZnO were observed in the XRD profile of untreated surface. Please explain.

Response 3: When zinc is etched with nitric acid, the reaction results in the formation of Zinc nitrate (Zn (NO3)2). Due to the water-soluble property of zinc nitrate, it can be easily rinsed away during the following cleaning step. As detailed in the washing step in section 2.1 sample preparation, the etched zinc was cleaned via ultrasonication for 10 min in 100% ethanol to remove residuals, via which the zinc nitrate can be simply rinsed away resulting in a clean and oxide-free zinc surface. This is confirmed by the XPS data in Figure 3 of the manuscript where there is only a single XPS peak for zinc without additional peaks at different binding energies. Regarding the ZnO diffraction peak on the untreated surface, this can be attributed to the fact that Zinc is a highly reactive metal and can react with the oxygen in the air to form a thin layer of zinc oxide on the surface, even under normal conditions 7. The manuscript has been revised: lines 310 to 312 of page 8.

Point 4: There are some format errors. For instance, line 128 “HNO3” should be “HNO3”; The unit of WCA should be “degree”; line 237 “H+”; line 238 “H2O”.

Response 4: The format errors have been corrected in the manuscript: line 128 of page 3 changing HNO3 to HNO3, line 165 of page 4 changing 107 to 107, line 238 of page 6 changing H+ to H+, line 239 of page 6 changing H2O to H2O

REFERENCE

  1. Ennaceri, H.; Wang, L.;  Erfurt, D.;  Riedel, W.;  Mangalgiri, G.;  Khaldoun, A.;  El Kenz, A.;  Benyoussef, A.; Ennaoui, A., Water-resistant surfaces using zinc oxide structured nanorod arrays with switchable wetting property. Surface and Coatings Technology 2016, 299, 169-176.
  2. Pesika, N. S.; Hu, Z.;  Stebe, K. J.; Searson, P. C., Quenching of growth of ZnO nanoparticles by adsorption of octanethiol. The Journal of Physical Chemistry B 2002, 106 (28), 6985-6990.
  3. Li, W.-J.; Shi, E.-W.;  Zhong, W.-Z.; Yin, Z.-W., Growth mechanism and growth habit of oxide crystals. Journal of crystal growth 1999, 203 (1-2), 186-196.
  4. Schierbaum, K.; Fischer, S.;  Torquemada, M.;  De Segovia, J.;  Roman, E.; Martin-Gago, J., The interaction of Pt with TiO2 (110) surfaces: a comparative XPS, UPS, ISS, and ESD study. Surface science 1996, 345 (3), 261-273.
  5. Watts, J. F.; Wolstenholme, J., An introduction to surface analysis by XPS and AES. John Wiley & Sons: 2019.
  6. Newbury*, D. E.; Ritchie, N. W., Is scanning electron microscopy/energy dispersive X‐ray spectrometry (SEM/EDS) quantitative? Scanning 2013, 35 (3), 141-168.
  7. Hammer, G.; Shemenski, R., The oxidation of zinc in air studied by XPS and AES. Journal of Vacuum Science & Technology A: Vacuum, Surfaces, and Films 1983, 1 (2), 1026-1028.

Round 2

Reviewer 1 Report

The authors addressed my requests. 

Reviewer 3 Report

In my opinion, this papaer is acceptable.